# Limitation in the Performance of Fine Powder Separation in a Turbo Air Classifier

Mohamed Abohelwa [1,*], Bernd Benker [2], Mehran Javadi [1], Annett Wollmann [1] and Alfred P. Weber [1]

1 Institute of Particle Technology, Clausthal University of Technology, 38678 Clausthal-Zellerfeld, Germany; mehran.javadi@tu-clausthal.de (M.J.); annett.wollmann@tu-clausthal.de (A.W); weber@mvt.tu-clausthal.de (A.P.W.)
2 CUTEC Forschungszentrum, 38678 Clausthal-Zellerfeld, Germany; bernd.benker@cutec.de
* Correspondence: ma96@tu-clausthal.de

**Abstract:** The deflector wheel classifier is a widely used device for the separation of fine powders in different industrial applications. The primary objective of the separation process is to achieve high-quality separation of fine powders characterized by a narrow particle size distribution and high separation sharpness. Theoretically, the reduction in the cut size is accomplished by decreasing the gas flow rate or increasing the rotational speed of the classifier, which amplifies the centrifugal forces compared to the drag forces exerted on the particles. This behavior is, indeed, observed in many cases, but it cannot be extrapolated arbitrarily. At their performance limit, classifiers may, against expectation, show an increase in cut size and, in addition, a reduction in the sharpness of the separation process. The limitation in the reduction in the cut size and in the improvement in the separation sharpness arises due to an imbalance between the operating rotational speed and flow rate, which results in a non-uniform flow field in the classifier. If the balance conditions are fulfilled, an optimum separation with a high separation sharpness can be achieved. In this work, CFD simulations validated by some experimental results are employed to represent this limitation, which is obtained by varying the operating parameters using different material densities with particles ranging from one to ten microns.

**Keywords:** air classifier; fine powders; CFD; cut size; separation sharpness

## 1. Introduction

The demand for finer particles has grown consistently over the past few years in the industrial sector. Fine particles are in high demand as fillers and fire retardants, as well as their use in the chemical, pharmaceutical, coal, and food industries [1,2]. For applications in the automotive industry as a coating [3] or even as a toner material in printers [4], high separation sharpness is required to reduce the amount of oversized particles in the fine fraction. By using different air classifiers and separating systems, it is possible to modify the particle size distribution to meet the increased requirements of the intended applications. Currently, liquid-crystal display (LCD) glass with very precise particle distributions is produced in the television and cell phone industries [2]. Gravity classifiers typically reach cut sizes from 200 μm to the millimeter range, while centrifugal classifiers may achieve cut sizes of 1 μm to 200 μm [5].

For the classification of particles with a size of 1 μm, it is crucial to generate and maintain an appropriate flow field. Generally, spiral air classifiers with free vortex flow and deflector wheel classifiers with forced vortex flow may be applicable [6].

Producing finer materials with smaller cut sizes can be achieved by altering both the geometric and operational parameters [7,8]. When the classifier geometry is fixed, the cut size solely depends on the operating parameters, such as the volume flow rate and rotational speed of the classifier blades. Particle classification in the deflector wheel classifier depends on the balance of two forces acting on the particles: the drag force, $F_d$,

and the centrifugal force, $F_c$. Coarse particles are transported to the outer space of the classifier because the centrifugal force is dominant for them. Conversely, fine particles migrate with the mainstream airflow to the center of the classifier because of the dominant drag force.

From a force balance, it can be deduced that the cut size, $x_{50,th}$, is proportional to the square root of the radial velocity, $v_r$, and inversely proportional to the circumferential velocity, $v_\phi$ (cf. Section 2).

Therefore, for classifications with low cut size, high circumferential velocities, $v_\phi$, in combination with low radial velocities, $v_r$, are required. The circumferential velocity of the flow is proportional to the rotational speed of the blades of the classifier, while the radial velocity is proportional to the volume flow rate of the air [9].

In summary, theoretically, the production of finer particles can be achieved by high rotational speeds of the classifier blades and low volume flow rate. Indeed, Refs. [10–12] reported in studies on turbo air classifiers that the cut size decreased monotonically as the rotational speed of the air classifier increased. At the same time, Ref. [10] observed a decrease in the separation sharpness with increasing rotational speed, which was confirmed by [9]. References [13,14] investigated the impact of operating parameters, including the air inlet velocity and the rotor speed, on the cut size of an air classifier and found a monotonic reduction in the cut size when reducing the air inlet velocity or increasing the rotational speed. In addition, Ref. [14] reported that the powder feed rate had a minor impact compared to the effects of the rotor speed and air inlet velocity in their experiments on a cage-type separator. A monotonic reduction in the cut size as a result of increasing the rotational speed has also been found by [6,15].

While most studies indicate a monotonic decrease in the cut size of particles with increasing rotational speed, Ref. [16] demonstrated an unconventional outcome. When classifying particles with an average size of 63 μm, they reached a minimum cut size at 120 rpm with increasing rotor speed but observed an increase in the cut size by further increasing the rotational speed. Repetitions of the experiment at higher particle concentrations led to the same trend.

Figure 1 summarizes the progression of the cut size with rotational speed observed in various studies, namely, [6,10,13,15,16]. Since these studies employed different circumferential velocities, $v\phi_r$, and operated at different cut sizes, the velocities shown in the figure were multiplied by a factor for comparison, and the cut sizes were normalized to the highest measured value.

While the commonly observed trend of decreasing separation limits with increasing rotational speed can be explained by Equation (1), the rebound of $x_{50}$ found by [16] remains surprising for the time being. One starting point could be the particle movement between the blades. Reference [17] has shown, for example, that, in the case of large particles, which enter the inter-blade space with low circumferential velocity, they only reach the circumferential velocity necessary for separation in the centrifugal field after colliding with the blades. However, also the particles used by [16] were relatively large ($x_{mean} = 63$ μm). Therefore, the question remains open whether an increase in the cut size is also observed and relevant for the separation of smaller particles (e.g., smaller than 10 μm).

In addition to the cut size, the separation sharpness is also affected by speed and flow rate, albeit in a slightly different way; while Ref. [16] does not address the evolution of the sharpness of cut at different operating parameters, several publications e.g., Refs. [6,9] provide some indication of the requirement for high sharpness values.

Low sharpness of separation results from an imbalance between the operating rotor speed and volume flow rate. This imbalance can be explained by the significant disparity between the rotational speed of the blades and the rotational speed of the airflow within the rotor region. As a result of the imbalance, a smooth, impact-free flow between the blades is impeded, leading to the formation of undesired vortices between the blades that decrease the separation sharpness by causing the back mixing of fine and coarse particles. An optimum separation sharpness is assumed to be obtained at operating parameters that

realize the balance in the form of a low difference between the rotational speeds of the flow and the blades. For each rotational speed, there would then be an optimum flow rate that realizes this balance, and vice versa.

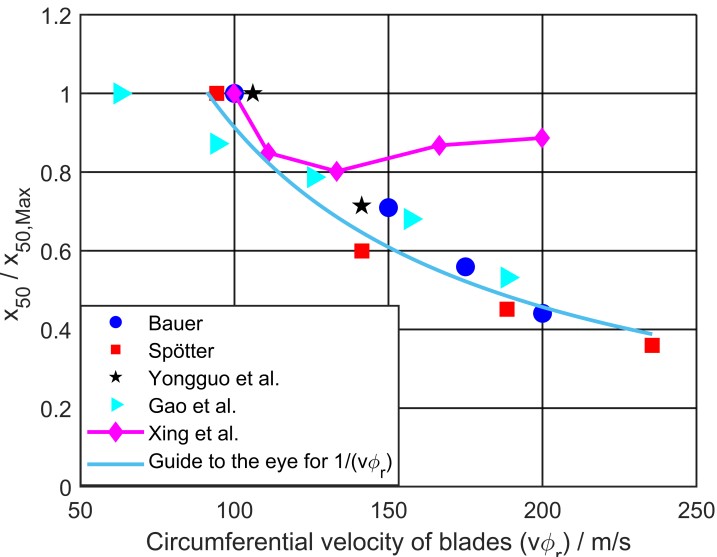

**Figure 1.** Trend of cut size evaluation due to variation in blade rotational speeds. For better comparison, $v\phi_r$ was scaled with a certain factor and $x_{50}$ was normalized: For [6], $v\phi_r = v\phi_r' \times 1$. For [15], $v\phi_r = v\phi_r' \times 6$. For [13], $v\phi_r = v\phi_r' \times 8.5$. For [10], $v\phi_r = v\phi_r' \times 15$. For [16], $v\phi_r = v\phi_r' \times 53$.

This study examines the separation characteristics of ultra-fine powders (<5 µm), focusing specifically on cut size and separation sharpness. In order to achieve this objective, the flow dynamics between the classifier blades are modeled using computational fluid dynamics (CFD), and the separation curve is obtained using a particle tracking methodology. The validity of the simulation findings is assessed by comparing them to the experimental data.

## 2. Methodology

In the results section, the experimentally obtained trends of cut size and separation sharpness of a deflector wheel classifier are combined with the trends calculated by CFD and particle tracking. The structure of the Results section is primarily based on the parameters that affect the cut size. These include the radial velocity, the circumferential velocity, the radius on which the particle separation is located, and the material density (see Equation (1)). With the complex flow in a deflector classifier, it is evident that a change in the circumferential velocity (rotational speed) influences the radial velocity (volumetric flow) and vice versa. However, the consequences of each velocity are discussed separately. First, an evaluation of experimental and CFD results is conducted, followed by a presentation and discussion of the effect of rotational speed on the cut size and separation sharpness for two volume flows. In addition, the trend of the theoretical cut size calculated according to Equation (1) is shown for two positions: the inner and outer radius. The aim of this approach is to evaluate the applicability of the basic equation for the calculation of the cut size in the ultra-fine particle range. Furthermore, addressed is the question of whether the separation occurs at the inner radius or, as is commonly assumed, at the outer radius. In addition, a material with a different density is considered. Next, the effect of the volume flow rate for two speeds on the cut size and separation sharpness is discussed. Assuming that Equation (1) is valid, the extent of the cut size may be determined analytically. However, it is significantly more challenging to estimate the separation sharpness. Consequently, the concluding section of the results discusses criteria for achieving high sharpness, relying on the connection between circumferential and radial velocity discussed in earlier sections.

Considering the two forces' equilibrium (drag and centrifugal forces) and assuming that particle motion is in the Stokes range, the theoretical cut size, $x_{50,th}$, can be calculated using Equation (1) [18], where $v_r(r)$ is the radial velocity of the flow, $v_\phi(r)$ is the circumferential velocity of the flow, $\eta$ is the dynamic viscosity of the air, $\rho_p$ is the density of the particle, and $r$ is the radius of the classifier, where the classification occurs.

$$x_{50,th} = \sqrt{\frac{18 \cdot v_r \cdot r \cdot \eta}{v_\phi^2 \cdot \rho_p}} \tag{1}$$

For the relevant radius, $r$, often the outer tip of the blades is used, although this could sometimes not precisely predict the actual cut size of the separation.

To assess the separation process, the separation curve or tromp curve can be considered, in which the separation efficiency, $T(x)$, is plotted against the particle diameter. The separation efficiency, $T(x)$, represents the percentage of feed material that is present in the coarse material fraction after the separation process and can be calculated from Equation (2). From the separation curve, the cut size, $x_{50}$, is determined at a separation efficiency of 50%. In Equation (2), $m_c$ represents the mass of coarse material, $m_{feed}$ is the mass of feed material, $q_c$ is the particle density distribution of the coarse material, and $q_{feed}$ is the particle density distribution of the feed material:

$$T(x) = \frac{m_c}{m_{feed}} \cdot \frac{q_c(x)}{q_{feed}(x)} \cdot 100\% \tag{2}$$

To calculate the separation efficiency from CFD simulations and produce the simulation's separation curves, each particle is tracked from the aerosol inlet into the classifier until it reaches its final position in the coarse or fine tubes. The input parameters required for Equation (2) are calculated at various positions, including the inlet (feed), fine material outlet, and coarse material outlet. By applying Equation (2), the separation percentage is then calculated for each particle diameter.

The separation sharpness, $k$, is determined from the ratio of particle sizes at the separation efficiencies of 25% ($x_{25}$) and 75% ($x_{75}$), as given in Equation (3) [19].

$$k = \frac{x_{25}}{x_{75}} \tag{3}$$

The effect of particle agglomeration, often observed in real classification experiments, is not considered in the CFD simulations. In the experimental classification process, the feed material may not be fully dispersed, resulting in agglomerated fine particles acting as coarse particles, which may be present in the coarse fraction after classification. As a result, $T(x)$ may not drop to zero, and a division ratio, $\tau$ (distance to the zero line), is present. However, in the CFD simulation, the feed particles are fully dispersed, resulting in no division ratio in the CFD results. To enable comparison, the division ratio of the experimental results, resulting from incomplete particle dispersion, is not considered, and the separation curves of the experimental results are corrected, as represented by Equation (4) and described by [6,20]. Here, $T(x)$ represents the corrected separation efficiency, while $T^*(x)$ represents the measured separation efficiency without correction. This also implies that for valid comparison, only low particle concentrations are allowed in the experiments with a mass loading of approximately 1%.

$$T(x) = \frac{T^*(x) - \tau}{1 - \tau} \tag{4}$$

## 3. Simulation and Experimental Methods

The classifier used in this study is a homemade horizontal deflector wheel classifier with a vertical axis of rotation.

The classifier features 16 airflow inlet nozzles distributed around the periphery, arranged in a manner that is tangential to the circumference at the tip of the blades.

Furthermore, 4 of the 16 total inlets are used to feed particles carried by the airflow (aerosol inlets), while the remaining 12 are used for particle dispersion. The 24 blades of the classifier have an inner radius of 60 mm, an outer radius of 100 mm, and a height of 40 mm. During the classifying process, coarse particles are transported to the outer area of the classifier and discharged through four separate pipes. Fine particles are transported to the center of the classifier and exit through a tube located at the center of the rotor blades. A dry disperser (RODOS) from Sympatec is used for pre-dispersing and conveying the feed material.

The powder used for this study is Saxolith 2, and the particle diameter distribution in the experiments was measured using a HELOS laser diffraction spectrometer from Sympatec (Figure 2).

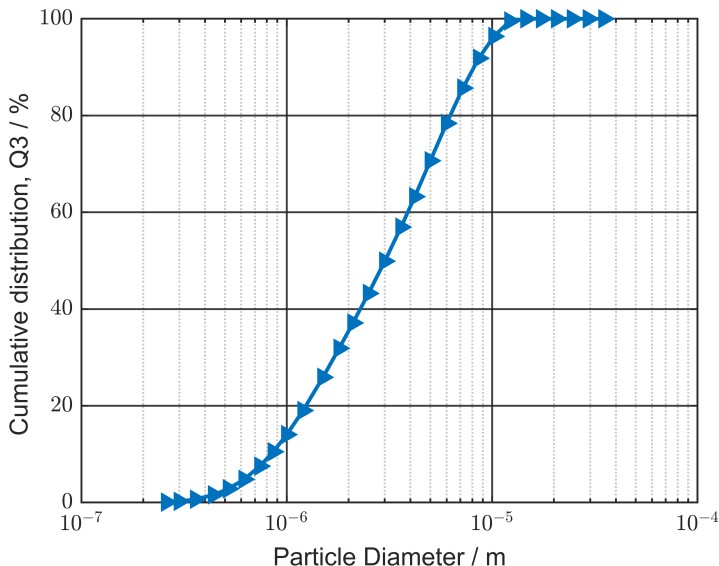

**Figure 2.** Particle size distribution of the Saxolith 2 powder.

Figure 3 shows a 3D representation of the deflector wheel classifier, and Table 1 summarizes all basic parameters of the classifier.

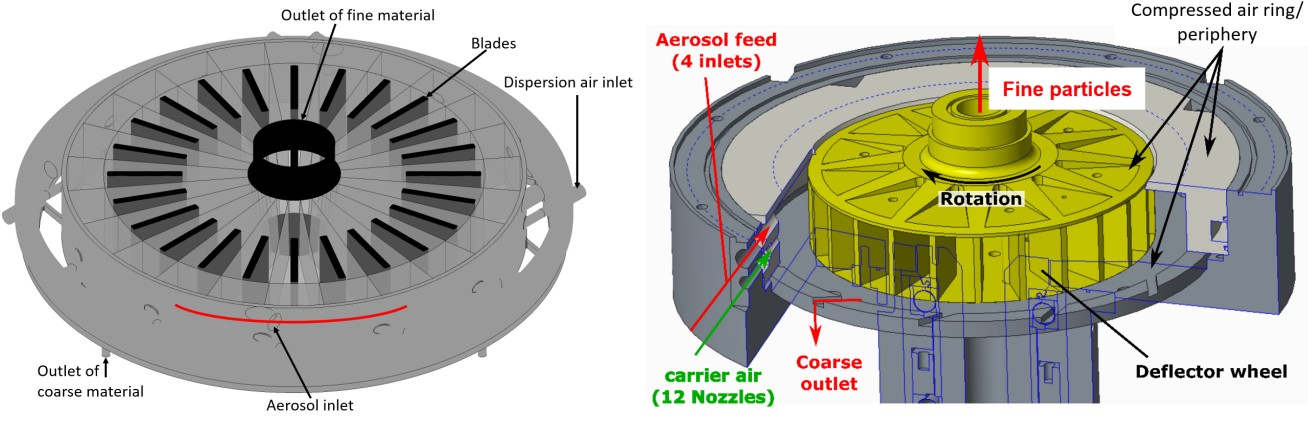

(**a**) Isometric view of CFD geometry          (**b**) View of the classifier as in reality

**Figure 3.** A 3D representation of the classifier in CFD (**a**) and in reality (**b**).

**Table 1.** Basic parameters of the classifier.

| Number of Blades | Outer Blade Radius | Inner Blade Radius | Height of the Classifier | Number of Aerosol Inlets | Number of Dispersion Air Inlets |
|---|---|---|---|---|---|
| 24 | 100 mm | 60 mm | 40 mm | 4 | 12 |

The primary method of investigation is computational fluid dynamics (CFD) simulations that utilize the Euler–Lagrange particle tracking model. The classifier geometry used in the simulations, as shown in Figure 3a, accurately represents the classifier used in the experiments, as depicted in Figure 3b. Some of the CFD results were validated against experimental data obtained under identical operating conditions. In the simulation, changes are implemented to the particle density, while particle size distribution is maintained according to the original material. Consequently, the impact of the air volume flow rate and classifier rotational speed on the minimum achievable cut size is discussed.

The CFD simulations in this study were performed using a commercial CFD solver ANSYS CFX 19.2, which solves the Reynolds-Averaged Navier–Stokes (RANS) equations. For these simulations, the Menter shear-stress transport (SST) turbulence model was used [21], which is a combination of two-equation turbulence models ($k - \varepsilon$ and $k - \omega$) and is adequate for predicting flow behavior near boundaries (walls) and far away in the bulk flow [22]. To solve the dispersed phase, the discrete phase model (DPM) was employed, which analyzes particle behavior from both Lagrangian and discrete viewpoints. The Lagrangian perspective differs from the Eulerian viewpoint in that the former analyzes fluid behavior through particle tracking, while in the latter, the conservation equations are formulated based on the concept of control volume [23].

The treatment of the interface between the rotating blades and the adjacent stationary parts is achieved by using the multiple reference frame model (MRF), also known as the frozen rotor Generalized Grid Interface (GGI) approach. In this approach, the rotating parts are kept stationary (frozen), and the centrifugal and Coriolis forces are included to account for the rotation. The rotating system is subjected to the no-slip condition. The frozen rotor approach omits the non-stationary effects resulting from the rotation of the blades and the interaction between the rotating blades and the stationary parts. However, it is a robust approach compared to others, such as the mixing plane approach (also known as the stage approach), which may not accurately predict loading and is unsuitable for applications with strong component coupling or severe wake interaction effects. The frozen rotor approach does not require high computing capacity [22]. This approach has been employed in earlier studies and has proven to be satisfactory for such applications, including those described by [22,24,25].

The boundary conditions for the aerosol inlet and the fine and coarse material outlets are given as mass flow rates, which were determined experimentally. To ensure the robustness of the simulation, the boundary condition at the inlet of the dispersion flow is set to a static pressure of 1 bar, which balances the total inlet and outlet mass flow rates.

The flow rates in the simulations correspond to those used in the experiments. In the simulations, the particles are introduced as a discrete diameter distribution at the inlet of the classifier, based on the HELOS measurements.

## 4. Results

In this section, the effect of operating parameters on the classification process, such as the rotational speed of the blades and the volume flow rate, is discussed. To investigate the effect of rotational speed variations, the volume flow rate is kept constant at 72 m$^3$/h and 108 m$^3$/h and the rotational speed is varied from 3500 rpm to 10,000 rpm. Furthermore, to investigate the effect of volume flow rate, the rotational speed of the classifier is kept at 5000 rpm, and the volume flow rate is varied from 36 m$^3$/h to 108 m$^3$/h. Finally, criteria will be given to obtain a high separation sharpness.

### 4.1. Comparison between CFD, Experimental, and Theoretical Results

Figures 4 and 5 illustrate the cut size and separation sharpness obtained from both the CFD simulations and experimental results. The data are presented for a constant volume flow rate of 72 m$^3$/h and various rotational speeds (Figure 4), as well as a rotational speed of 5000 rpm and various flow rates (Figure 5). Regarding the trend of cut size and separation sharpness in response to variations in operating parameters, both the CFD and experimental results agree well and show a similar trend, albeit with slightly different absolute values. The consistency between the CFD approach and experimental data validates the CFD procedure utilized in this study. As discussed in Sections 4.2 and 4.3, this validation procedure allows additional investigations that may be extended to other operating characteristics. Appendices A and B contain the separation curves obtained from CFD simulations and experimental measurements. Figure 4 indicates that as the rotational speed increases, the cut size is reduced until a minimum achievable cut size is reached (at 6500 rpm). Further increasing the rotational speed does not lead to a further decrease in the cut size. Instead, increasing the rotational speed reduces the separation sharpness. Similar to Figure 4, Figure 5 demonstrates that once the cut size reaches a certain value (at 54 m$^3$/h), decreasing the volume flow rate does not result in a reduction in the cut size but rather a decrease in separation sharpness.

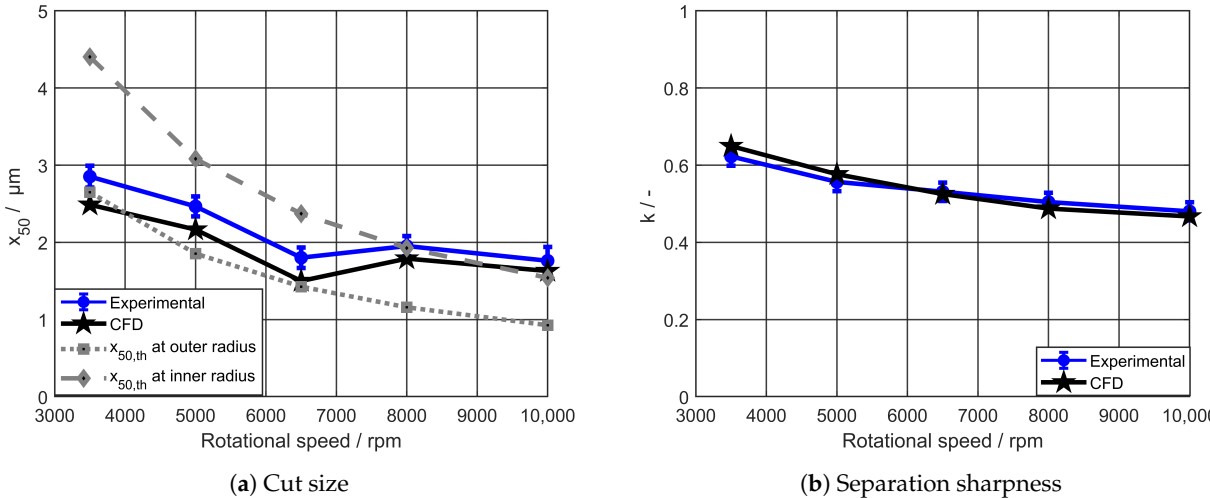

(**a**) Cut size  (**b**) Separation sharpness

**Figure 4.** Cut size (**a**) and separation sharpness (**b**) obtained from CFD and experimental results alongside with the theoretical cut size calculated from Equation (1) at the outer blade radius ($r_o$ = 0.1 m) and at the inner blade radius ($r_i$ = 0.06 m) at different rotational speeds and a constant flow rate of 72 m$^3$/h, particle density = 2750 kg/m$^3$ (bars indicate the scatter of experimental results).

Theoretical cut sizes, $x_{50,th}$, calculated using Equation (1), are also plotted in Figures 4 and 5 at both the outer blade radius ($r_o$ = 0.1 m) and the inner blade radius ($r_i$ = 0.06 m). The comparison between the cut sizes obtained from CFD simulations and experimental results reveals a good agreement with $x_{50,th}, r_o$ (i.e., referring to the outer radius) when operating at low rotational speeds up to 6500 rpm or high volume flow rates above 54 m$^3$/h. However, at high rotational speeds or low flow rates, where a limited cut size is reached, the actual cut size deviates from $x_{50,th}, r_o$ (i.e., referring to the outer radius) and becomes closer to $x_{50,th}, r_i$ (i.e., referring to the inner radius). This shift in cut size is attributed to the non-uniform flow and the formation of vortices between the blades at these flow conditions, which play a crucial role in particle separation, as discussed in Sections 4.2 and 4.3. The flow vortices entrain coarser particles toward the inner radius of the blades, altering the equilibrium and leading to the observed shift. These findings highlight the challenge in selecting an appropriate radius, $r$, in Equation (1), as this depends on the specific operating conditions and the complexity of the flow.

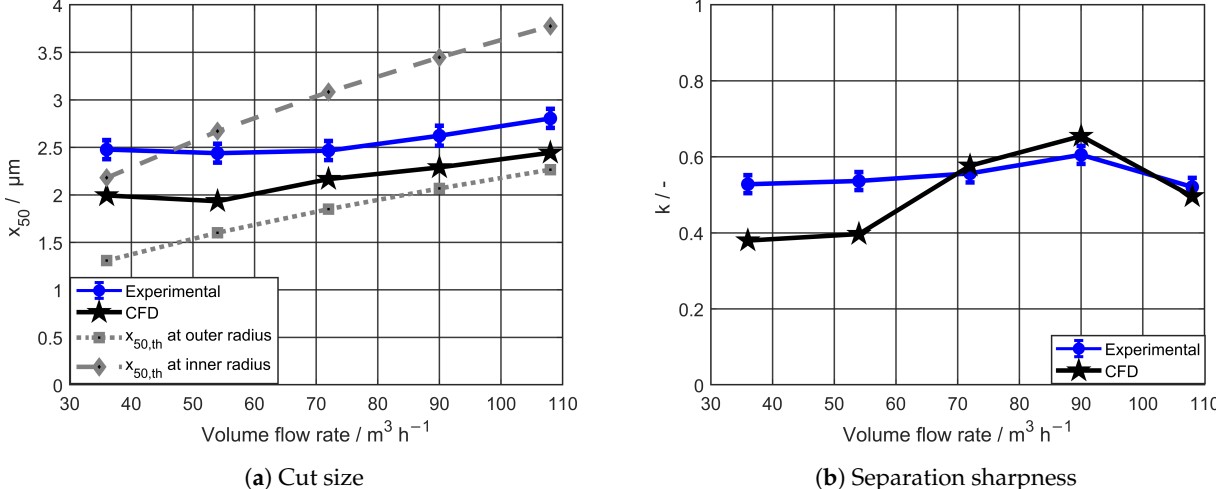

(**a**) Cut size    (**b**) Separation sharpness

**Figure 5.** Cut size (**a**) and separation sharpness (**b**) obtained from CFD and experimental results alongside with the theoretical cut size calculated from Equation (1) at the outer blade radius ($r_o$ = 0.1 m) and at the inner blade radius ($r_i$ = 0.06 m) at different flow rates and a constant rotational speed of 5000 rpm, particle density = 2750 kg/m$^3$ (bars indicate the scatter of experimental results).

### 4.2. Effect of Blades Rotational Speed on the Classification Process

In order to comprehensively understand this limiting phenomenon, further investigations are performed using CFD simulations at different particle densities and operating parameters.

In Figure 6a, the cut size is plotted against the rotational speed for different flow rates and particle densities. The four curves exhibit a similar trend to each other and to those obtained from the experiments, where the cut size is monotonically decreased by increasing the rotational speed until it reaches a minimum value. A further increase in the rotational speed does not lead to a smaller cut size. Figure 6b indicates that a further increase in the rotational speed beyond this limit could decrease the separation sharpness instead. The smallest cut size is achieved at a rotational speed of 6500 rpm for all cases, although the value of the smallest achievable cut size varies among the cases. This variation illustrates the coupled dependence of the separation cut size on the rotational speed, flow rate, and particle density. The cut sizes for higher-density material are commonly smaller compared to lower-density material. The trend of the separation sharpness does not follow the same pattern for all cases as the cut size. In the two cases with a flow rate of 72 m$^3$/h, the separation sharpness decreased monotonically with increasing rotational speed, consistent with previous findings [11]. However, for a flow rate of 108 m$^3$/h, the sharpness peaked at 6500 rpm, where the smallest cut size was achieved. This observed peak in separation sharpness is attributed to the fulfillment of balance conditions between the rotational speed and the flow rate at this specific operating point. This balance leads to the highest separation sharpness among all other operating conditions.

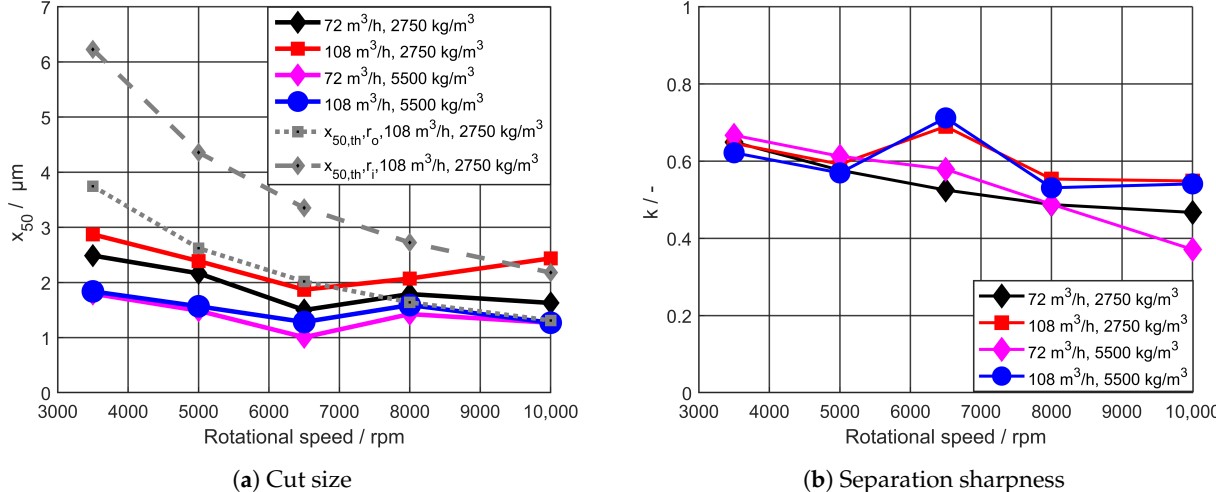

(**a**) Cut size              (**b**) Separation sharpness

**Figure 6.** Cut size (**a**) and separation sharpness (**b**) obtained from CFD results at different rotational speeds.

For a comprehensive understanding of this behavior, it is essential to observe the flow behavior within the classifier. Figure 7 displays the distribution of radial flow velocities on a horizontal plane parallel to the nozzles of the aerosol inlet (parallel to the red curve in Figure 3a). The plot includes cases with a flow rate of 72 m$^3$/h and varying rotational speeds. For a representative illustration of the flow behavior, this plane has been chosen, however, the flow on other planes from the bottom to the top of the classifier shows similar patterns.

As the rotational speed is increased, the flow becomes less homogenous, and the width of the range of radial velocities becomes wider, which causes the back mixing of coarse and fine particles [26]. Through the formed flow vortices from this flow non-uniformity, a negative impact on the separation process can be obtained.

The formation of flow vortices between the blades can present a challenge in reducing the cut size of the separation process. These vortices have the potential to entrain coarse particles towards the center of the classifier and subsequently collect them as fine materials, primarily through highly negative radial velocities on the blade pressure side. As a consequence, this phenomenon can increase the cut size, despite the high centrifugal forces generated from the blade rotation at high rotational speeds. The same could happen for fine materials to be entrained with the flow vortices toward the outer space of the classifier and to be collected as coarse material, thereby reducing the separation sharpness.

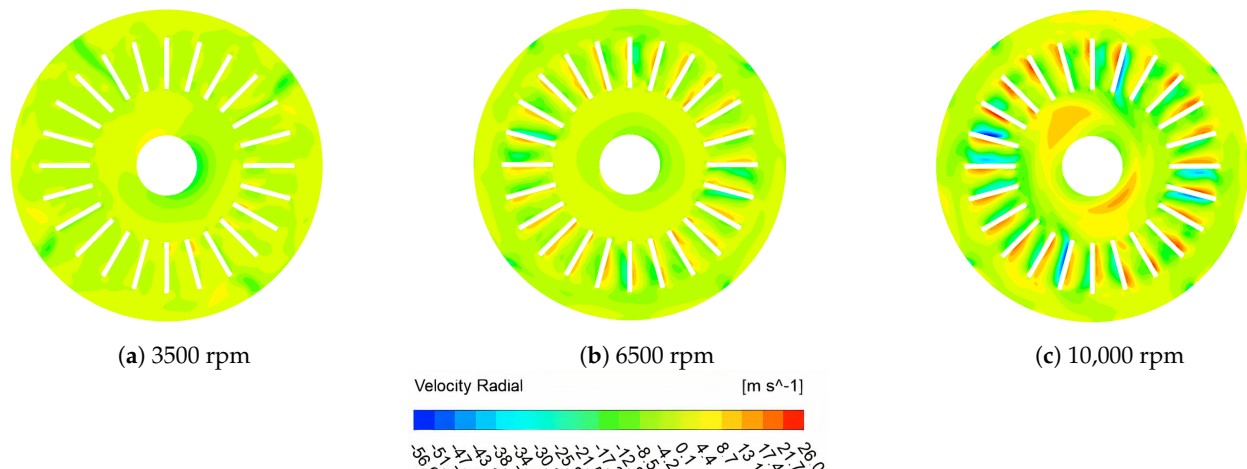

(**a**) 3500 rpm        (**b**) 6500 rpm        (**c**) 10,000 rpm

**Figure 7.** Radial velocity contours for the cases operating at 72 m$^3$/h and different blade rotational speeds at a horizontal plane parallel to the nozzles of the aerosol inlet, density = 2750 kg/m$^3$.

Figure 8 displays the radial velocity distribution of the flow for cases operating at a flow rate of 108 m³/h and different blade rotational speeds. Similar to Figure 7, these figures illustrate the impact of flow uniformity on the separation process.

The enhanced homogeneity of the flow between the blades for the case operating at 6500 rpm can be attributed to the balance achieved between the flow rate and the blade rotational speed, as explained further in Section 4.4. This balance results in the lowest cut size and the highest separation sharpness compared to other cases (cf. Figure 6). At this specific operating condition, flow vortices between the blades are minimized, and the centrifugal forces generated from the blade rotation become more crucial for the separation. This reduces the possibility of coarse particle entrainment with the airflow to the center of the classifier and the back mixing, resulting in a lower cut size and a higher separation sharpness. Moreover, the rotational speed not only affects the circumferential velocity of the flow but also changes the distribution of the radial velocity (similarly stated in [24]), which is found to be more responsible for the separation performance compared to the circumferential velocities as obtained at high rotational speeds.

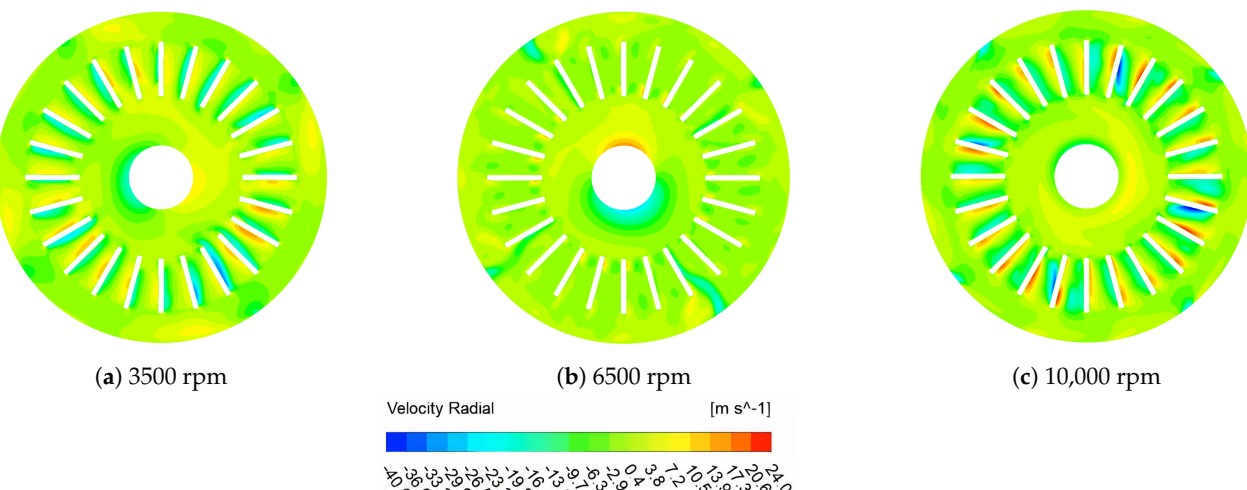

(**a**) 3500 rpm　　　　　　　　　　(**b**) 6500 rpm　　　　　　　　　　(**c**) 10,000 rpm

**Figure 8.** Radial velocity contours for the cases operating at 108 m³/h and different blade rotational speeds at a horizontal plane parallel to the nozzles of the aerosol inlet, density = 2750 kg/m³.

### 4.3. Effect of the Volume Flow Rate on the Classification Process

Figure 9a represents the cut size obtained from CFD simulations for two different particle densities by varying the flow rate. As the flow rate decreases, the cut size decreases monotonically to a minimum value at a flow rate of 54 m³/h. A further reduction in the cut size is not possible after the classification reaches this limitation. Figure 9b shows, however, that reducing the flow rate below 90 m³/h monotonically reduces the separation sharpness.

Reference [5] stated that higher flow rates are required for a sharp cut. From Figure 9b, it can be seen that this is not always the case as the separation sharpness peaks at 90 m³/h and at a higher flow rate of 108 m³/h, the sharpness drops. This also indicates that at a constant rotational speed, an optimum flow rate for the sharpest cut exists.

Based on Equation (1), the cut size for both particle densities should be related to each other, as expressed in Equation (5). In Figure 9a, the cut size for particle density, $\rho_p = 2750 \text{ kg/m}^3$, is scaled using Equation (5). The scaled $x_{50}$ demonstrates good agreement with the course of $x_{50}$ for particle density, $\rho_p = 5500 \text{ kg/m}^3$, obtained from CFD simulations.

$$x_{50,2} = x_{50,1} * \sqrt{\rho_{p,1}/\rho_{p,2}} \tag{5}$$

From Figures 6 and 9, it can be concluded that lower cut sizes are not always obtained at higher rotational speeds or lower volume flow rates. Conversely, higher rotational speeds and lower flow rates could have a negative effect on the separation sharpness. To achieve

an efficient separation, a balance between the flow rate and the rotational speed of the classifier should be maintained.

The impact of variation in flow rate on the cut size is less significant compared to that of rotational speed. However, it has a more significant impact on the separation sharpness. This might explain why, in Figure 6a, the lowest cut size at a rotational speed of 6500 rpm is the same for all cases, despite differences in flow rates, but with a peak in the separation sharpness for the cases operating at 108 m$^3$/h.

In order to underline this conclusion, the contour plots in Figure 10 illustrate the radial velocities for cases operating at 5000 rpm and varying flow rates. A reduction in volume flow rate below the optimum value causes an imbalance with the blade rotational speeds, resulting in the formation of flow vortices between the blades. The plots clearly show that the case operating at 90 hlm$^3$/h has the most uniform flow between the blades, leading to the highest separation sharpness. Conversely, at other flow rates (e.g., case operating at 36 hlm$^3$/h), a wide range of radial flow velocities is evident, with more likely negative flow velocities. Highly negative and positive flow velocities (toward the center and the exterior of the classifier, respectively) are responsible for the back mixing of coarse and fine particles, resulting in low separation sharpness.

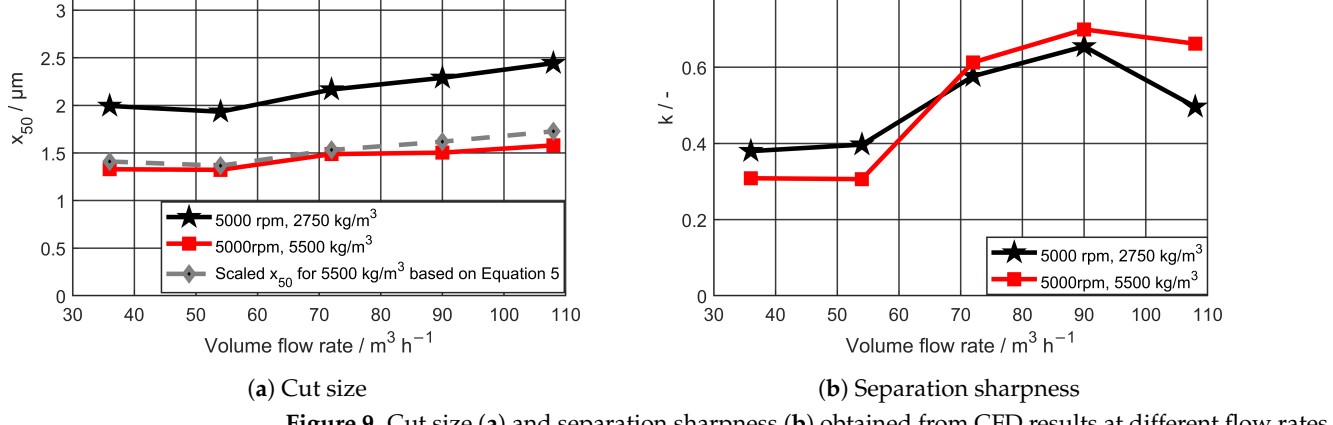

(**a**) Cut size

(**b**) Separation sharpness

**Figure 9.** Cut size (**a**) and separation sharpness (**b**) obtained from CFD results at different flow rates.

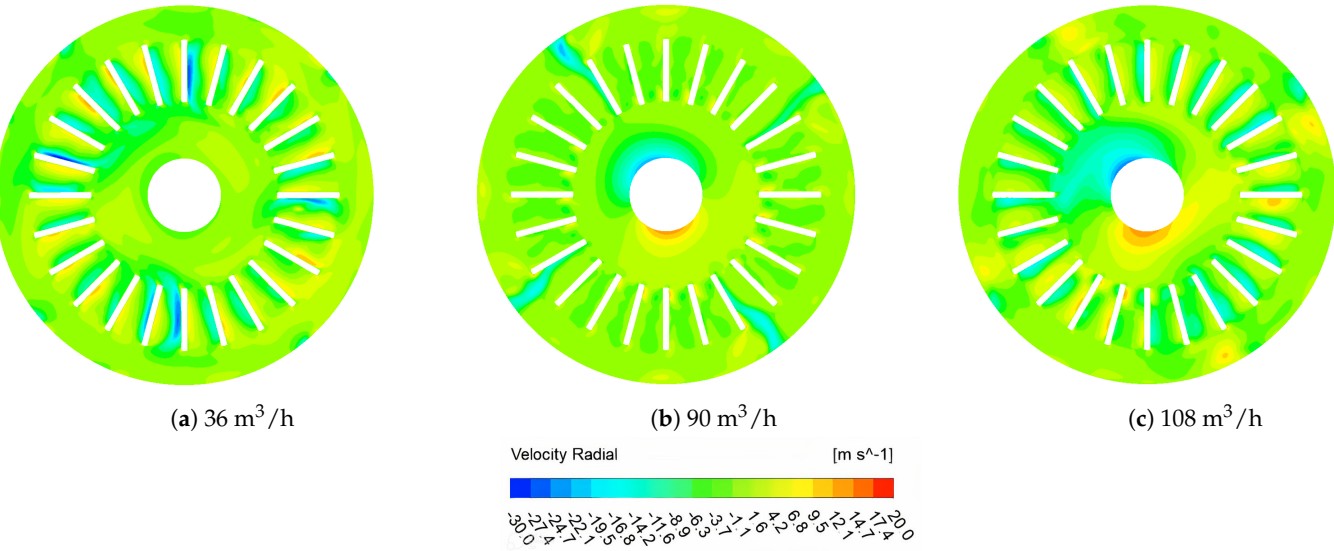

(**a**) 36 m$^3$/h

(**b**) 90 m$^3$/h

(**c**) 108 m$^3$/h

**Figure 10.** Radial velocity contours for the cases operating at 5000 rpm and different volume flow rates at a horizontal plane parallel to the nozzles of the aerosol inlet, density = 2750 kg/m$^3$.

### 4.4. Criteria for High Separation Sharpness

References [27,28] stated in their works that the circumferential velocity of the flow should be equal to that of the rotor for a uniform flow distribution between the blades. This may result in an impact-free flow between the blades and prevent the formation of flow vortices in the rotor blade channels. Figures 11 and 12 show the relative circumferential velocity of air ($v\phi_{rel} = v\phi_a - v\phi_r$), which is the difference between the circumferential velocity of the airflow (in a stationary frame of reference) ($v\phi_a$) and the rotor rotational speed ($v\phi_r$) evaluated from the blades leading edge ($r = 0.1$ m) to trailing edge ($r = 0.06$ m). Each point is calculated as an average over a circular cylinder expanded from the bottom to the top of the classifier at the corresponding diameter ($r$). The distribution of the relative circumferential velocity ($v\phi_{rel}$) may explain the process of vortex formation between the blades as shown in the following.

For a flow rate of 72 m$^3$/h (cf. Figure 11a), the course of $v\phi_{rel}$ is close to zero for 3500 rpm but deviates increasingly more from zero for higher speeds. This explains the reduction in the separation sharpness in response to increased rotational speeds (cf. Figure 6b). A similar behavior is observed at a higher flow rate of 108 m$^3$/h (Figure 11b). Here, however, the most pronounced approach of $v\phi_{rel}$ to the zero line is found for 6500 rpm. As can be seen in Figure 8 from the radial velocities, for the case of 6500 rpm, a small deviation between rotor circumferential velocity and tangential air velocity is associated with the least formation of vortices, i.e., with the most homogeneous flow field between the blades. This leads also to the highest sharpness of cut as confirmed in Figure 6b. A similar correlation is found for the variation of the flow rate. Figure 12 shows the relative circumferential velocity $v\phi_{rel}$ versus blade radius for different flow rates at a constant rotational speed of 5000 rpm. Again, the highest sharpness of cut is observed for the smallest deviation from the zero line (see Figure 10b). Therefore, it can be concluded that the more uniform the flow field, the higher the sharpness of cut. The imbalance between the blade rotational speed and the volume flow rate can now be understood in terms of high $|v\phi_{rel}|$ leading to vortex formation.

To achieve classification with high separation sharpness and a uniform flow, the rotational speed and the volume flow rate should be balanced with each other, and low values of $|v\phi_{rel}|$ between the blades should be obtained. For a fixed rotational speed, there exists an optimum flow rate, and vice versa. Choosing the optimum operating parameters is difficult to predict as the circumferential velocity of the flow between the blades depends on both parameters, the flow rate and the rotational speed, among other parameters, such as the design of the blades and the particle mass concentration. To overcome this problem, the blade design should be optimized to have better interaction with the flow and prevent vortex formation. Therefore, further research is required to develop smart blade designs.

Altering the rotational speed of the blades or the volume flow rate, while keeping other parameters constant, leads to changes in the distribution of both radial and circumferential velocities of the airflow. Consequently, predicting the cut size using Equation (1) becomes challenging when the flow becomes non-uniform at the operating conditions, where the limitation is reached. However, it has been shown that the application of Equation (1) with $r_i$ and $r_o$ give reasonable limits for the effective separation grain $x_{50}$. More robust conclusions can be drawn with respect to the sharpness of cut. It is clearly found that if the balance condition is fulfilled at the outer radius of the blades, i.e., $v\phi_{rel} \approx 0$, the highest sharpness of cut is obtained.

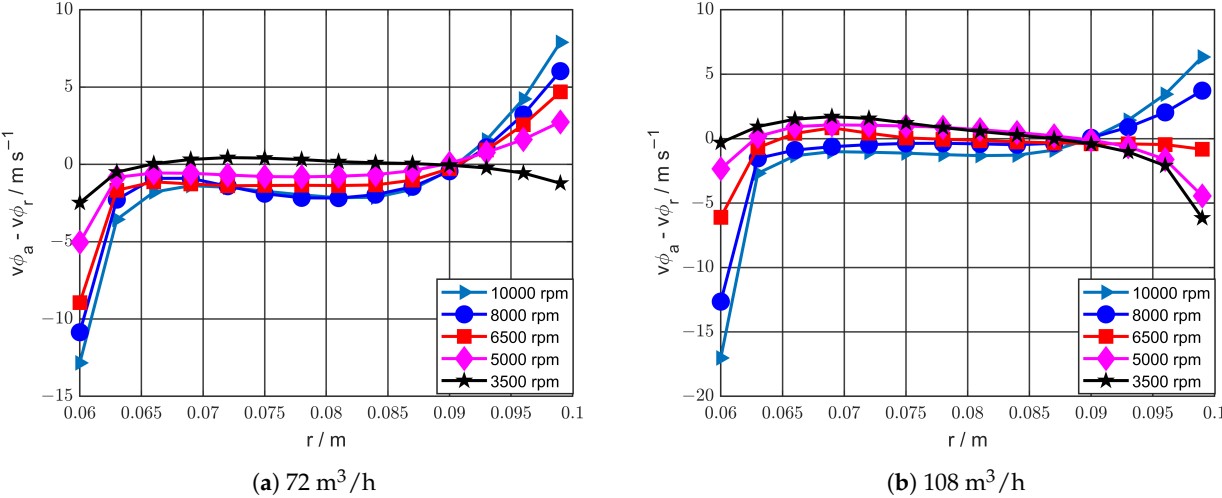

(**a**) 72 m$^3$/h

(**b**) 108 m$^3$/h

**Figure 11.** Radial dependence of average relative circumferential velocity between the air and the rotor blades from leading to trailing edge of the blades at two constant flow rates of 72 m$^3$/h (**a**) and 108 m$^3$/h (**b**).

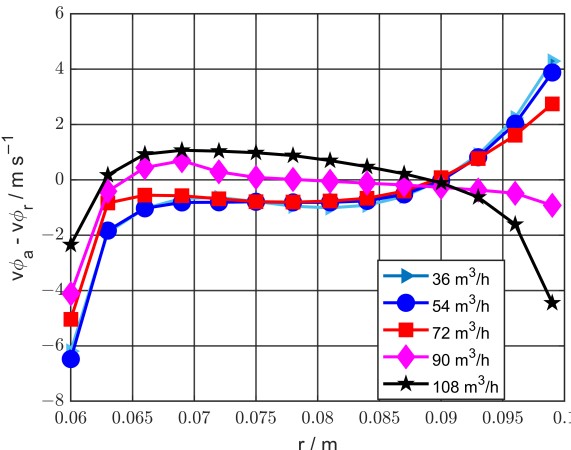

**Figure 12.** Radial dependence of average relative circumferential velocity between the air and the rotor blades from leading to trailing edge of the blades at a constant rotational speed of 5000 rpm.

## 5. Conclusions

In this study, the effects of diverse operational parameters on separation sharpness and cut size within a deflector wheel classifier were analyzed, employing both numerical simulations and experimental observations. The key findings are summarized as follows:

1. Strong agreement between numerical simulations and experiments validated model simplifications in the CFD simulations, supporting the use of simplifications such as the frozen rotor approach.
2. The simulated flow field was used to investigate the origin of the limitations of achievable cut size and separation sharpness.
3. The cut size exhibited an anticipated reduction with increasing classifier wheel revolution rate, which is consistent with existing literature. Yet, at a certain point, cut size began to rise again due to vortex formation between the blades.
4. At low rotational speeds and high volume flow rates, using the outer blade radius provides a good estimate of the cut size.
5. Effective equilibrium of circumferential speeds between blades and carrier gas yielded high separation sharpness, which is in agreement with earlier works. This balance allowed for separation sharpness enhancements, even with simple blade configurations.

6. A further improvement in the separation sharpness may be obtained by optimizing the geometry and the number of blades, which is the goal of ongoing work.

**Author Contributions:** Conceptualization, M.A., A.W. and A.P.W.; methodology, M.A.; software, M.A.; validation, M.A., A.W. and B.B.; formal analysis, M.A. and M.J.; investigation, M.A.; resources, M.A. and B.B.; data curation, M.A. and M.J.; writing—original draft preparation, M.A.; writing—review and editing, A.P.W., A.W., B.B. and M.J.; visualization, M.A.; supervision, A.P.W.; project administration, A.P.W.; funding acquisition, A.P.W. All authors have read and agreed to the published version of the manuscript.

**Funding:** This research was funded by the German Research Foundation (DFG) (grant number: WE 2331/22-2) within the Priority Program 2045. The publishing fee of the article was funded by the Open-Access Publishing Fund of Clausthal University of Technology.

**Data Availability Statement:** The data are available upon reasonable request from the corresponding author.

**Acknowledgments:** We express our gratitude to Leonard Hansen for manufacturing the classifier used in this study. We acknowledge financial support by Open Access Publishing Fund of Clausthal University of Technology.

**Conflicts of Interest:** The authors declare no conflict of interest. The funders had no role in the design of the study; in the collection, analyses, or interpretation of the data; in the writing of the manuscript; or in the decision to publish the results.

## Nomenclature

| | |
|---|---|
| $\rho_p$ | Density of the particle [kg/m$^3$] |
| $k$ | Separation sharpness [%] |
| $m_c$ | Mass of coarse material [kg] |
| $m_{feed}$ | Mass of feed material [kg] |
| $q_c$ | Particle density distribution of the coarse material [$\mu$m$^{-1}$] |
| $q_{feed}$ | Particle density distribution of the feed material [$\mu$m$^{-1}$] |
| $q_f$ | Particle density distribution of the fine material [$\mu$m$^{-1}$] |
| $T(x)$ | Separation efficiency [%] |
| $v_\phi$ | Circumferential velocity [m/s] |
| $v_r$ | Radial velocity [m/s] |
| $x_{25}$ | Particle size at separation efficiecncy = 25% [$\mu$m] |
| $x_{75}$ | Particle size at separation efficiecncy = 75% [$\mu$m] |
| $\eta$ | Dynamic viscosity of the air [Pa·s] |
| $v\phi_a$ | Circumferential velocity in stationary frame for air [m/s] |
| $v\phi_{rel}$ | Relative circumferential velocity between air and rotor [m/s] |
| $v\phi_r$ | Rotor rotational speed [m/s] |

## Appendix A. Separation Curves Obtained from CFD Simulations and Experimental Measurements at a Constant Flow Rate but Different Rotational Speeds

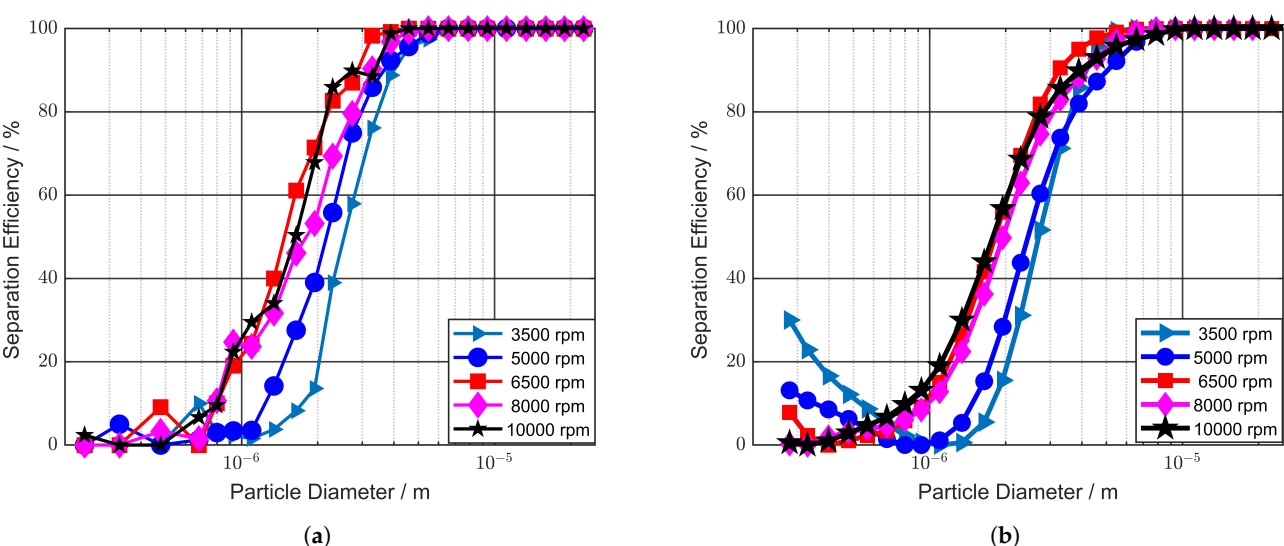

(**a**)    (**b**)

**Figure A1.** Separation curves at a constant volume flow rate (72 m$^3$/h) with a monotonic variation in the classifier blades rotational speed, particle density = 2750 kg/m$^3$, (**a**) CFD, (**b**) experimental.

## Appendix B. Separation Curves Obtained from CFD Simulations and Experimental Measurements at a Constant Rotational Speed but Different Flow Rates

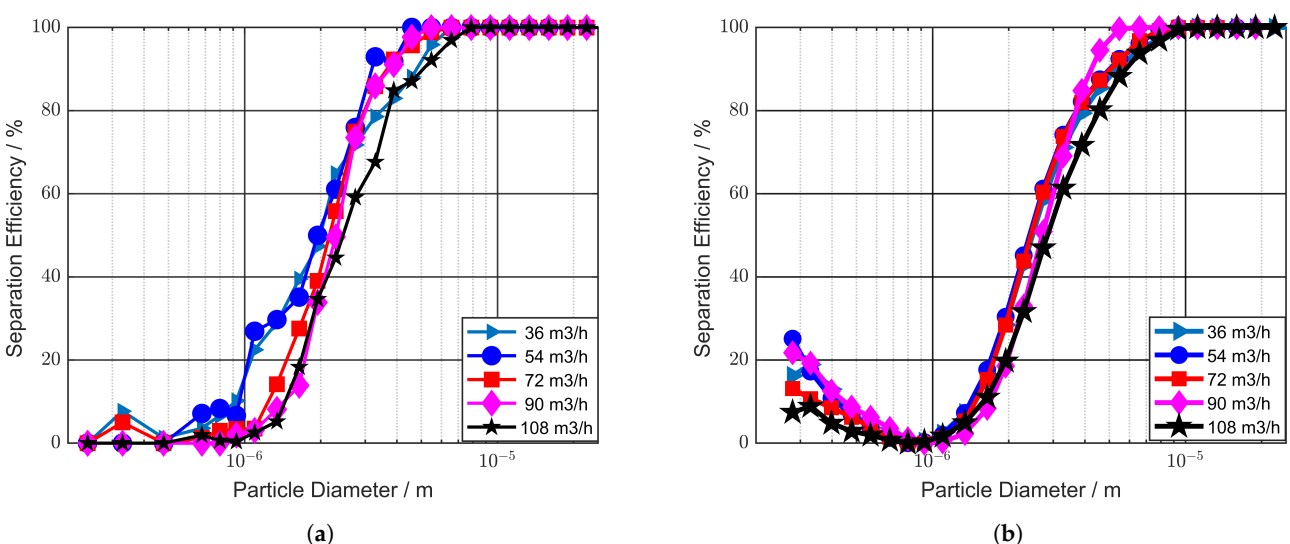

(**a**)    (**b**)

**Figure A2.** Separation curves at a constant blade rotational speed (5000 rpm) with a monotonic variation in the volume flow rate, particle density = 2750 kg/m$^3$, (**a**) CFD, (**b**) experimental.

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
