# Peer review of "Limitation in the Performance of Fine Powder Separation in a Turbo Air Classifier"

_processes, doi:10.3390/pr11102817_

Round 1

Reviewer 1 Report

Comments:

Overall, the authors have done a good job in the manuscript. I have the following comments:

·         The introduction provides an overview of the topic, which is good. However, it could benefit from some restructuring to make it more cohesive and easier to follow. Consider organizing the content into clear paragraphs, each addressing a specific aspect of the introduction, such as the importance of particle separation in various industries, the use of the deflector wheel classifier, the main objective of particle classification processes, the factors affecting the cut size, and the unconventional outcome observed in some studies.

·         The main objective of particle classification processes is mentioned in the introduction, but it could be made more explicit. Consider rephrasing and structuring the sentence to clearly state the objective of the study.

·         Towards the end of the introduction, there is a mention of "separation characteristics in the ultra-fine powder range." While it sets up the scope of the study, it may be helpful to elaborate further on what "ultra-fine powder range" means and why it is significant.

·         Minor corrections in English are required in the manuscript.

Minor Corrections in English are required. 

Author Response

Dear Reviewer,

thank you very much for the effort of reviewing and the suggestions. The annotations from all reviewers lead in the same direction and did show room for improvement in our paper. 

Please find below our changes according to your comments:

Regarding your comments:

  1. The introduction provides an overview of the topic, which is good. However, it could benefit from some restructuring to make it more cohesive and easier to follow. Consider organizing the content into clear paragraphs, each addressing a specific aspect of the introduction, such as the importance of particle separation in various industries, the use of the deflector wheel classifier, the main objective of particle classification processes, the factors affecting the cut size, and the unconventional outcome observed in some studies.

Reply:

Thank you very much for this comment. We have rewritten the introduction accounting for the comments of the reviewer to make it more cohesive and easier to follow. In this context, the motivation for the use of deflector wheel classifiers and the industrial fields of application have also been dealt with in more detail.

  1. The main objective of particle classification processes is mentioned in the introduction, but it could be made more explicit. Consider rephrasing and structuring the sentence to clearly state the objective of the study.

Reply:

Thank you very much for this comment. The main objective of particle classification was again deepened in several places. In particular, it was emphasized which factors determine the achievable lower separation limit and, above all, also the separation sharpness. For this purpose, the criteria for a separation with a high sharpness were again deepened in chapter 4.4.

  1. Towards the end of the introduction, there is a mention of "separation characteristics in the ultra-fine powder range." While it sets up the scope of the study, it may be helpful to elaborate further on what "ultra-fine powder range" means and why it is significant.

Reply:

Thank you very much for this comment. We have defined in more detail the size range of ultra-fine powders under consideration and justified why these powders in particular are of particular interest and in which industrial sectors they are increasingly used.

Lines 107-108:

This study examines the separation characteristics of ultra-fine powders (<5 μm), focusing specifically on cut size and separation sharpness.

Lines 18-25:

“The demand for finer particles has grown consistently over the past few years in the industrial sector. Fine particles are in high demand as fillers, fire retardants, and their use in the chemical, pharmaceutical, coal, and food industries ((Gao et al. [1]) and Hosokawa Alpine AG [2]). For applications in the automotive industry as a coating (Gu et al., [3]) or even as a toner material in printers (Parthasarathy [4]), high separation sharpness is required to reduce the amount of oversized particles in the fine fraction. Classifiers are also often combined with impact mills or jet mills operating in a dry atmosphere to remove the fines from the process and recycle the coarse material for regrinding (Koeninger et al.,  [5]).

  1. Minor corrections in English are required in the manuscript.

Reply:

Thank you very much for this comment. The English has been revised again throughout the manuscript and, where possible, simplified as well as improved in quality.

We hope that we have satisfactorily addressed all of your comments and would like to thank you again for your helpful input!

Reviewer 2 Report

Reorganize the entire structure of the manuscript.

Review all important related publications in the Introduction section.

Add a new section "Methodology" and list all the mathematical equations used.
The English writing has to be improved.

List the basic parameters of the studied turbo air classifier in a table.

Why does Figure 2 appear after Figure 3? Rearrange the figures in the manuscript.

All parameters in the equation must be explained in the text.

Conclusions must be clear and well organized.

Moderate editing of English language required

Author Response

Dear Reviewer,

thank you very much for the effort of reviewing and the suggestions. The annotations from all reviewers lead in the same direction and did show room for improvement in our paper. 

Please find below our changes according to your comments:

Regarding your comments:

  1. Reorganize the entire structure of the manuscript.

Reply:

Thank you very much for this comment. We have revised the whole structure of the manuscript with the aim of making the common thread clearer and the approach more comprehensible. The new structure is now:

  1. Introduction
  2. Simulation conditions and experimental method
  3. Methodology
  4. Results

    4.1. Comparison between CFD, experimental and theoretical results

    4.2. Effect of blades rotational speed on the classification process

    4.3. Effect of the volume flow rate on the classification process

    4.4. Criteria for high separation sharpness

  1. Conclusions

  1. Review all important related publications in the Introduction section.

Reply:

Thank you very much for this comment. The amount of cited literature in the introduction has been expanded to address all relevant publications as much as possible. In particular, the industrial areas of application for deflector wheel classifier have also been added again, e.g. Koeninger et al. [5].

  1. Add a new section "Methodology" and list all the mathematical equations used.

Reply:

Thank you very much for this comment. We have introduced a new Methodology section to clarify the approach and all the equations used are included within the manuscript.

The English writing has to be improved.

  1. The English writing has to be improved.

Reply:

Thank you very much for this comment. The English has been revised again throughout the manuscript and, where possible, simplified as well as improved in quality.

  1. List the basic parameters of the studied turbo air classifier in a table.

Reply:

Thank you very much for this comment. The basic parameters of the deflector wheel classifier have been described in the new Table 1 and in the text.

  1. Why does Figure 2 appear after Figure 3? Rearrange the figures in the manuscript.

Reply:

Thank you very much for this comment. This error has been corrected so that the sequence of figures now matches the text.

  1. All parameters in the equation must be explained in the text.

Reply:

Thank you very much for this comment. All parameters used are now described and further elaborated in the text.

  1. Conclusions must be clear and well organized.

Reply:

Thank you very much for this comment. The Conclusions has been completely rewritten to concisely and significantly summarize the findings.

Lines 400-418:

“The current investigation has analyzed numerically and experimentally the effects of diverse operational parameters, including the rotational velocity of the blades and flow rates, on the separation sharpness and cut size within a deflector wheel classifier. The good agreement between numerical and experimental results for the separation curves justified model simplifications in the CFD simulation such as the frozen-rotor approach. Therefore, the simulated flow field was used to investigate the origin of the limitations of achievable cut size and separation sharpness, respectively. As expected from a simple force balance, the cut size decreases with increasing revolution rate of the classifier wheel in agreement with most of the data from literature. For this operational range, using the outer blade radius provides a good estimate of the cut size. However, at a certain point the cut size increases again with further increasing revolution rate, effectively limiting the achievable lower cut size, which is attributed to the formation of vortices between the blades. In particular, these vortices also impair a good sharpness of the cut. Based on the flow simulation, it can be concluded that high separation sharpness is obtained when the circumferential speeds of the blades and of the carrier gas are balanced, which has been indicated in previous studies, e.g., by Bauer [7] and Galk [26]. By adjusting the rotational speed to the flow rate, even for simple straight short blades a sharpness of cut in the range of 0.7 was achieved. A further improvement of the separation sharpness may be obtained by optimizing the geometry and the number of the blades which is the goal of ongoing work.”

We hope that we have satisfactorily addressed all of your comments and would like to thank you again for your helpful input!

Round 2

Reviewer 2 Report

 1. Reorganize the structure of the manuscript.
Section1. Introduction
Section2. Methodology
Section3. Simulation conditions and experimental method
Section4. Results and Discussions
Section5. Conclusions

2  List all mathematical equations used in the section "Methodology", but in Section "Introduction".

 3. Line 125-127, Figure 2 and Figure 3 must follow the text, not in another page.

 4. Conclusions.  List all your new findings with bullet points (1...2...3...) . Do not cite any references from other publications in Section "Conclusions". That shall be shown in Section "Introduction"

Minor editing of English language required

Author Response

Dear reviewer,

Thank you very much for your constructive comments, which we have incorporated into the revised version of the manuscript and which have significantly improved the quality of the manuscript.

Regarding your comments:

  1. Reorganize the structure of the manuscript.
    Introduction
    Section2. Methodology
    Section3. Simulation conditions and experimental method
    Section4. Results and Discussions
    Section5. Conclusions

Reply:

Thank you very much for this comment. The structure of the manuscript has been reorganized according to your recommendations.

  1. List all mathematical equations used in the section "Methodology", but in Section "Introduction".

Reply:

Thank you very much for this comment. All mathematical equations used in the study have been listed in the Section "Methodology".

  1. Line 125-127, Figure 2 and Figure 3 must follow the text, not in another page.

Reply:

Thank you very much for this comment. The positions of the figures have been changed so that they now follow the text.

  1. List all your new findings with bullet points (1...2...3...) . Do not cite any references from other publications in Section "Conclusions". That shall be shown in Section "Introduction"

Reply:

Thank you very much for this comment. The new findings of the study have been listed with bullet points. The citations of the references have been removed from the Section "Conclusions".

  1. Minor editing of English language required

Reply:

The English has been revised again throughout the manuscript and, where possible, simplified as well as improved in quality.

Round 3

Reviewer 2 Report

The authors improved the manuscript according to my comments. It can be accepted after text editing revision.